# Characterising performance information use in the primary healthcare systems of El Salvador, Lebanon and Malawi: multiple qualitative case study protocol

Wolfgang Munar ![ORCID],[1] Syed Shabab Wahid,[2] Martha Makwero,[3,4] Fadi El-Jardali,[5] Luckson Dullie,[6] Wen-Chien Yang[1]

For numbered affiliations see end of article.

**Correspondence to**
Dr Wolfgang Munar;
wolfgangmunar@gwu.edu

## ABSTRACT

**Introduction** Governments in low-income and middle-income countries (LMICs) and official development assistance agencies use a variety of performance measurement and management approaches to improve the performance of healthcare systems. The effectiveness of such approaches is contingent on the extent to which managers and care providers use performance information. To date, major knowledge gaps exist about the contextual factors that contribute, or not, to performance information use by primary healthcare (PHC) decision-makers in LMICs. This study will address three research questions: (1) How do decision-makers use performance information, and for what purposes? (2) What are the contextual factors that influence the use or non-use of performance information? and (3) What are the proximal outcomes reported by PHC decision-makers from performance information use?

**Methods and analysis** We present the protocol of a theory-driven, qualitative study with a multiple case study design to be conducted in El Salvador, Lebanon and Malawi. Data sources include semi structured in-depth interviews and document review. Interviews will be conducted with approximately 60 respondents including PHC system decision-makers and providers. We follow an interdisciplinary theoretical framework that draws on health policy and systems research, public administration, organisational science and health service research. Data will be analysed using thematic analysis to explore how respondents use performance information or not, and for what purposes as well as barriers and facilitators of use.

**Ethics and dissemination** The ethical boards of the participating universities approved the protocol presented here. Study results will be disseminated through peer-reviewed journals and global health conferences.

## STRENGTHS AND LIMITATIONS OF THIS STUDY

⇒ Strengths include the use of theory to guide study design, data collection and reporting; the consideration of rival explanations; and the use of triangulation of data sources, respondent accounts and researcher interpretation.

⇒ The use of a case study design with embedded units in different country contexts can contribute to theoretical generalisations about the influence of contextual factors on performance information use and non-use, and thus influence future comparative research.

⇒ Limitations include reduced transferability of findings to contexts other than the three participating countries and to other populations of decision-makers and providers. The use of virtual interviews may create potential loss of rapport between interviewers and respondents.

and can contribute to the achievement of Universal Health Coverage in the era of Sustainable Development Goals. High-performing PHC systems have also proven to be key in the preparedness for and response to pandemics and other public health emergencies.[1 2]

PHC has been defined as a whole-of-government and whole-of-society approach that combines multisectoral policy and action, empowered people and communities and primary care and essential public health functions as the core of integrated health services.[3] PHC systems are first points of entry into health service delivery, are essential for people-centred service delivery and connect citizens to health systems.[4]

During the last 40 years, performance measurement and management (PMM) systems have become prevalent in healthcare management and organisation.[5–7] Governments, official development assistance agencies and various global health partnerships

## INTRODUCTION

This protocol aims to describe how decision-makers and providers in three low-income and middle-income countries (LMICs) use available data to assess the performance of their primary healthcare (PHC) systems. Acquiring this knowledge is important for improving PHC systems responsiveness

have used diverse PMM approaches to improve performance of policies and programmes in maternal and child health,[8] [9] HIV/AIDS, malaria and tuberculosis,[10] and other global health priorities. Outcomes-driven financing approaches have also been used as a means to improve PHC system performance.[11]

## PMM SYSTEMS

PMM systems were originally conceived as ensembles of management control mechanisms designed to stimulate the delivery of organisational priorities and influencing desirable organisational behaviours.[12–14] However, depending on contextual factors and historical antecedents, PMM systems have evolved in response to contrasting organisational logics.[15] *Directive* systems tend to be guided by a logic of consequences, are prevalent in systems that favour audit cultures,[16] are designed with a view towards accountability and follow the utility-maximising assumptions of *Homo economicus* in agency theory.[17] *Enabling* approaches are guided by logics of improvement and learning; can create conditions for adaptive and iterative cycles of error, reflection, sensemaking and corrective action; and conceive of performance as emergent processes, influenced by managers and workers' agency, motives, means and opportunities.[18] [19]

Studies on PMM systems' effectiveness have identified several sources of leverage for performance improvement in public sector organisations.[20–23] Organisational performance tends to be positively associated with PMM systems that reinforce workforce motivation[24]; promote performance measurement at multiple levels (ie, individual, interpersonal and interorganisational)[25]; and where decision-makers use the information generated by the PMM system .[26] [27]

Governments use, and official development assistance agencies promote, a diverse set of approaches to performance management including financial arrangements, accountability approaches and implementation strategies.[28] An evidence gap map of PMM interventions in the PHC systems of LMICs showed that most primary studies to date have focused on provider-level implementation strategies such as in-service training and supervision, and on financial arrangements like pay-for-performance.[29] The mapping exercise also identified absolute gaps in evidence for PMM interventions that operate at organisational levels, particularly accountability arrangements

like public release of performance information or social accountability. There is also limited knowledge about the role of contextual factors in enabling or hindering the use of performance information at the organisational level of teams, facilities and district health systems. Table 1 summarises the interventions mapped in the evidence gap map above.

The widespread use of PMM systems in the public sector, particularly in health, has shown that, when not tailored to context, PMM systems can not only be ineffective but can also contribute to negative outcomes such as gaming, goal displacement and data manipulation.[30] Further, public administration research has shown that decision-makers do not consistently use performance information and that, when they do, the largest impacts on service delivery are attained when it is used as part of organisational dialogues that inform changes in operational and strategic direction.[26] [27] [31] The literature has also shown that official development assistance agencies promote and use various PMM approaches for improving accountability to donors and beneficiaries; enhancing organisational learning and communications; and informing changes in strategic direction.[32]

The literature on routine health information systems (RHIS) in LMICs has identified organisational, behavioural and technical challenges to the production and use of information including, among others, fragmentation, duplication and poor data quality.[33] It has also been shown that even when quality health information is available, LMIC health managers may not use it, leading to suboptimal decision-making processes that may negatively affect governance and healthcare management. Previous research has also found that non-use of data from RHIS can be explained by lack of motivation or scarce capacity among decision-makers; and by non-existing or poorly functioning feedback and supervision mechanisms.[34–37]

To address the above gaps in evidence and increase the understanding of performance information use, or non-use, in LMIC settings, this article presents the protocol of a qualitative multiple case study.

## METHODS AND ANALYSIS
### Study aims and research questions
The study described here will assess the experiences of PHC decision-makers and providers with PMM systems in

---

**Table 1** Interventions and approaches in primary healthcare systems performance measurement and management

| Implementation strategies | Accountability arrangements | Financial arrangements |
|---|---|---|
| Provider-level: Clinical practice guidelines, reminders, in-service training and continuous education. Organisational-level: Clinical incident reporting; clinical practice guidelines; local opinion leaders; continuous quality improvement; and supervision. | Individual-level or organisational-level: Audit and feedback. Community-level: Public release of performance information, social accountability. | Individual-level and organisational level: Results-based financing, pay-for-performance and other provider incentives and rewards. |

---

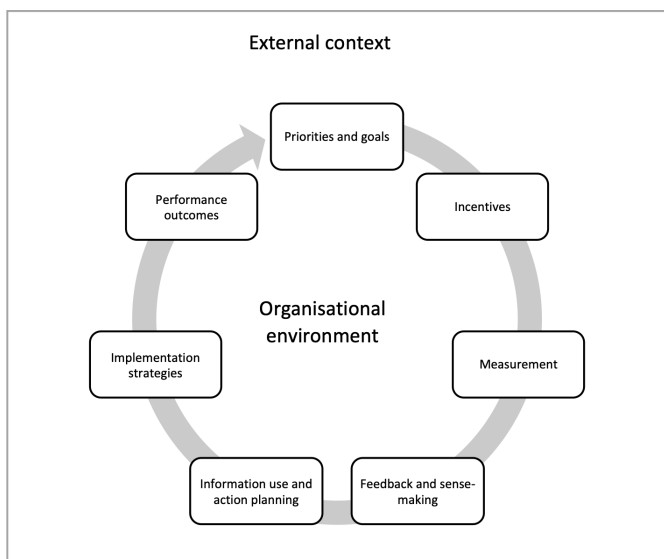

**Figure 1** Performance measurement and management system.

El Salvador, Lebanon and Malawi. Research findings will be used to inform an applied research agenda on PHC system performance; contribute to improve the measurement and management of PHC systems performance; and develop an evaluation framework for assessing performance information use in other country contexts. Our research questions are: (1) How do PHC system decision-makers use performance information, or not, and for what purposes? (2) What are the contextual factors that influence the decision to use performance information, or not? (3) What are the proximal outcomes reported by PHC decision-makers from performance information use and non-use?

## Theoretical framework

Based on PMM models in public administration research, implementation research and organisational science[6 22 38–40] we developed an interdisciplinary theoretical framework to help guide study design. PMM systems are conceptualised as continuous and recursive cycles of (1) organisational priorities and goals; (2) incentives; (3) performance measurement, feedback and sensemaking; (4) implementation strategies; and (5) performance outcomes,[6] as represented in figure 1.

Organisational priorities and goals are the ultimate expression of what desirable performance ought to be; they are identifiable in policy documents, summarised in logical models and sometimes reflected as measurable targets in performance frameworks. Incentive systems are managerial practices aimed at stimulating workforce motivation and fostering organisational performance by means of extrinsic and intrinsic stimuli. Extrinsic motivators include rewards, recognition, pay-for-performance, bonuses and in-kind incentives, among others.[41] Intrinsic motivators can trigger satisfaction of workers' basic psychological needs such as competence, autonomy, and

connection.[42 43] It is believed that both types of motivators are central to organisational performance.[44]

Performance measurement processes generate raw data about past performance and use metrics that reflect organisational priorities and goals. Performance data are usually compiled into registers that feed into RHIS, and can be summarised and disseminated via reports, scorecards and dashboards. Given the perceived low-quality of RHIS, particularly in LMICs,[35] performance data is also sometimes sourced from population surveys. The latter have become one of the most frequently used data sources for tracking health programmes' performance in LMICs.[36 37]

The data acquired via RHIS and/or population surveys are usually contrasted against expected targets and goals which, in turn, are disseminated in ways that generate performance information flows aimed at different users. Upward flows bring information through organisational hierarchies usually for reporting and accountability purposes. Information can also be fed back to the frontlines of service provision as part of feedback and audit, quality improvement or supportive supervision processes.[45] As organisational actors engage with performance data, ascribe meaning to it and imagine future courses of action in response to perceived gaps in performance, the managerial processes above can contribute to collective sensemaking,[46] a process that helps people 'understand issues or events that are novel, ambiguous, confusing or in some other way violate expectations'.[47] It can also inform decisions among organisational actors to engage or not in addressing the gaps in performance made evident by available information. Action plans, budgets, changes in service delivery and other processes of course-correction can then be considered for future implementation.

Once courses of action are decided, organisational actors can deploy various strategies to implement them. Implementation strategies help system actors appraise and respond in adaptive fashion to factors in their immediate environment that can enable or hinder collective action (see table 1). In the short-term, performance information can be used for planning, compliance, reporting or rapid course-correction purposes, among others; it can also be misused through gaming processes, or not used.[30] As iterative PMM cycles are repeated through time, performance information can also be used (or not) as the basis for testing new processes and services, for internal advocacy and/or for policy formation or redesign.

PMM cycles can contribute to proximal performance outcomes that feed into long causal chains of outcomes occurring at multiple levels within an organisation (eg, at individual, team and organisational levels). Outcomes can include (1) proximal changes resulting from using performance information (or not), such as action plans implemented, compliance with procedural standards, timely reporting and rapid course-correction; (2) intermediate effects emerge at the organisational behaviour level, and may include changes in workforce motivation,

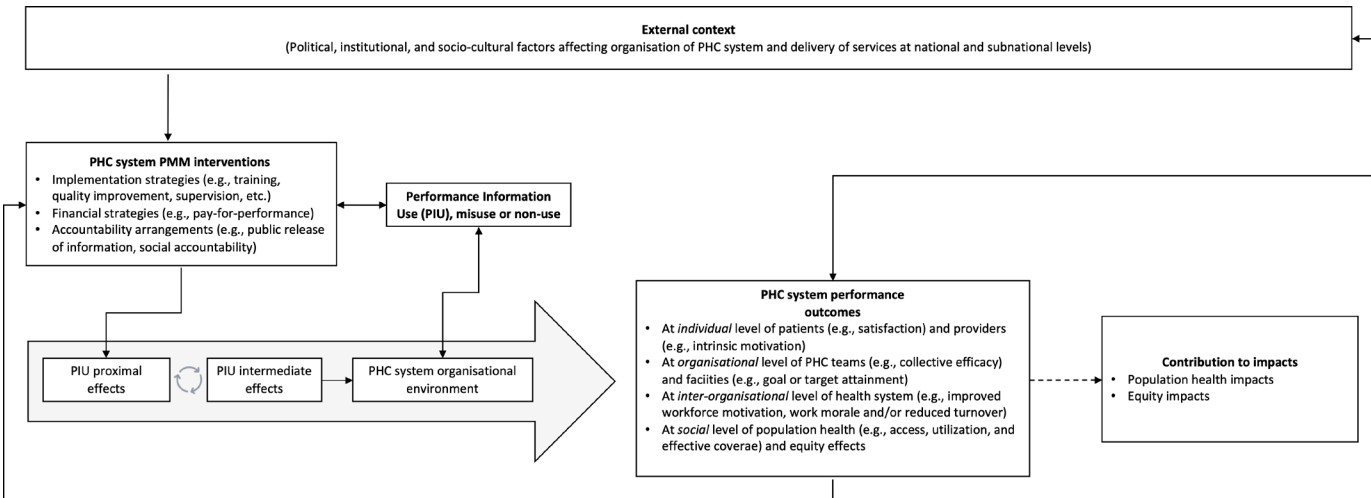

**Figure 2** Theoretical framework. PHC, primary healthcare; PMM, performance measurement and management.

job satisfaction, morale or organisational commitment; and (3) downstream population-level health effects and equity outcomes resulting from the iterative repetition of PMM cycles in dynamic and changing environments.

We integrated the elements of the PMM model described above into a theoretical framework that represents the hypothetical process of performance change and the role played by performance information use and non-use. The framework contains the following elements: external context; PMM approaches in use within the public sector; performance information production and use; PHC systems' internal organisational environment; and the causal pathways connecting performance information use and non-use to proximal, intermediate and distal outcomes.

The theoretical framework is represented in figure 2. Here, the managerial practices used to measure and change performance are influenced by external and internal contextual factors and by the implementation strategies in use and modulated by the use and non-use of performance information. The processes of change thus generated can contribute, via long causal chains, to a variety of outcomes and impacts. Proximal effects from performance information use are represented by single-loop learning effects[48] such as changes in planned action, rapid course-correction and improvements in service quality. The repetition of such iterative cycles may, in turn, contribute to the emergence of second-loop learning effects such as changes in strategic direction and new practices among service providers and managers.[49]

The use of performance information is causally linked to proximal performance outcomes at the individual level of providers and patients; those outcomes are also causally connected to intermediate outcomes at the organisational level such as improved workforce motivation, enhanced organisational commitment, increased trust between providers and PHC system users and reduced staff turnover, among others. These outcomes can contribute to distal population health and equity outcomes (intended and otherwise). Depending on context, the causal chain of outcomes described above can also be interrupted, be limited to isolated pockets of excellence, or be altogether absent.

## Study design
The present study will explore the uses of performance information in the PHC systems in El Salvador, Lebanon and Malawi. Investigation across different contexts allows for the generation of context-specific insights of value to local actors and, potentially, to broader understandings of the phenomena of interest.[50]

To address the research questions, we chose a theory informed, multiple case study design with embedded units of analysis.[51] Case studies are well-suited for obtaining an in-depth understanding of context-specific processes in complex systems.[52] Here, a case is defined as each country's PMM practices; the two units of analysis included are PHC service provision and PHC policy implementation at national and subnational levels.

## Study setting
### El Salvador
El Salvador is a lower-middle income country with a population of 6.4 million. Since the conclusion in 1992 of a civil war, the country reduced inequality by about 5% points between 2007 and 2016; increased coverage of institutional deliveries and immunisation to 98% and 93%, respectively; and achieved the under-5 mortality reduction for the Millennium Development Goals.[53 54]

Starting in 2009, El Salvador universalised access to free, comprehensive PHC. Existing infrastructure was reorganised into PHC networks, one for each of the departments in which the country is administratively divided. Service delivery was delegated to multi professional teams of PHC providers. The oversight of each departments' network is the responsibility of a decentralised Ministry of Health (MOH) coordination team called SEBASI in its Spanish acronym. PHC teams have a nominal catchment area of 3000 individuals and are co-located within the communities they serve. A basic PHC team is made up of one

medical doctor, two nurses and up to three community health promoters; some teams have specialised care providers. PHC teams provide community outreach as well as facility-based services and deliver a package of benefits containing approximately 300 interventions.[55]

In 2011, the government of El Salvador joined the Salud Mesoamerica Initiative (SMI), a public–private partnership focused on improving the performance of PHC systems in the eight nation states of Mesoamerica. In El Salvador, SMI operates in 75 PHC teams operating in the poorest rural municipalities in the country. PMM interventions used include PHC team target-setting; monitoring of PHC teams' performance using population and facility surveys and RHIS; provision of feedback to teams; and team-based in-kind incentives.[55 56]

### Lebanon

Lebanon is home to approximately 6.8 million people and is classified as an upper-middle income country.[57] However, the financial crisis that started in 2019 reduced real per-capita gross domestic product 37.1% between 2018 and 2021. The country also hosts the largest number of refugees per capita in the world[58] and has suffered additional internal shocks. The combined effect of these various shocks has put major pressure on an already stretched healthcare system.[59 60]

PHC services in Lebanon are provided by a combination of private-for-profit and not-for-profit providers; the latter are the most accessible and used sources of care by vulnerable Lebanese and refugee populations.[61 62] Lebanon's official PHC network is comprised of 213 centres that have contractual agreements with the Ministry of Public Health based on pre-met community care delivery standards.

In terms of performance measurement at the PHC level, the MOH has developed policies and practices to monitor service delivery patterns, quality of care and performance of PHC centres within the national network.[63] Monitoring involves regular visits by MOH inspectors and administration of patient satisfaction surveys.[63] Accreditation is also used to regulate the quality of care at the PHC level. By establishing a National Accreditation Program for PHC centres in 2009, the MOH aimed to ensure continuous and sustainable quality control, improve compliance with legal and safety standards, enhance transparency and accountability and establish a positive image of standards of practice and service at PHC centres.[63]

Despite the various health reforms implemented in Lebanon, there is still no active national strategic plan designed around PHC.[64–67] Furthermore, many PHC centres remain underdeveloped with no availability of basic diagnostic imaging and laboratory medicine, resulting in perceived lack of confidence in the quality of services offered.[60]

### Malawi

Malawi is a landlocked, low-income country with a population of approximately 18.6 million. The economy is mainly dependent on the agricultural sector which employs 80% of the population. A 5-year development plan, Malawi's Growth and Development Strategy, guides the country's development; the current plan is focused on education, health, agriculture, energy and tourism.[68]

Malawi's epidemiological profile combines a high burden of disease from both preventable conditions as well as non-communicable diseases. The country has a high population density and a total fertility rate of 4.4. Prevalent social determinants of health include poverty and inequality, high levels of illiteracy and limited coverage of social safety programmes.[68 69]

Primary care is the main platform for the delivery of health services in Malawi. However, the PHC system is characterised by poor distribution of human and physical resources, fragmentation of services and chronic shortages of staff.[70] To reduce service fragmentation, Malawi developed in 2017 a new community health policy centred on a team-based approach. Community health teams comprise health surveillance assistants, clinicians, environmental health officers and community health volunteers.[71]

### Data collection

The proposed study will use document review and semi-structured interviews with informants who typically hold 'great knowledge…[and] who can shed light on the inquiry issues'.[72] We will use document review to identify domestic priorities and explore the external context, available resources and ongoing official development assistance programmes. Documents to be reviewed include MOH policy documents, strategic frameworks, operational plans, results frameworks, performance reports and logical models, among others. Semi-structured interviews will be conducted with PHC decision-makers at the national and subnational levels and with PHC providers. To be eligible for inclusion, decision-makers will be current or former officials responsible for PHC system policy formulation or implementation at national and subnational levels; providers will be staff currently working as clinical care providers or community health workers.

Respondent selection and recruitment will follow an information power approach based on criteria that are suitable for reaching saturation in qualitative studies using non-probabilistic, purposive sampling.[73] We will design respondent sampling guided by our understanding of the types of participants that can provide highly specific information to address the study's research questions; insights from the preliminary theoretical framework; and responsive to the quality of the dialogue elicited during data collection. The estimated number of respondents to be interviewed in the three countries is approximately 20 respondents per site, for a total of 60 respondents. However, sampling numbers will be further refined, and may be expanded, based on preliminary analysis of data as data collection is ongoing. Respondent inclusion criteria will be calibrated to the context of each study

setting; site-specific approaches to data collection will be reported in each country case study.

In the interviews with service providers, we will explore experiences about the PHC system organisational environment; the ways in which PHC performance is measured, analysed and made sense of; the extent to which performance information is used or not, and for what purposes, and the reported effects from using performance information. Interviews with decision-makers at national and subnational levels will explore PHC priorities, goals and/or targets; characterise the public sector institutional context; explore sources and frequency of performance data appraisal; and inquire about the uses of performance information. We will also triangulate the data resulting from document review and interviews, and the experiences reported PHC by the two types of respondents.

In each country, the research team will develop a Project Brief summarising the study's aims and highlighting the voluntary nature of participation. An invitation to participate in the interview will be sent individually via email to each potential respondent. Once the respondent agrees to participate in the interview, a remote interview will be scheduled (or in-person, if allowed by an ethical review board). Before initiation of the interview, the interviewer shall read the consent form and obtain verbal consent from the interviewee which will be recorded and reflected in the interview transcript accordingly. Site-specific interview guidelines are available in online supplemental file 1.

### Analysis

Interviews will be audio taped, transcribed verbatim and imported into NVivo V.12.0. Transcripts will be coded independently by at least two researchers in each country. We will use an iterative, directed approach to analysis[74] informed by the theoretical framework. The latter shall also inform the design of a codebook to guide deductive coding of the data. Inductive codes emerging from the data will also be identified and included in the analysis. We will convene analytical workshops among the research teams in participating countries to discuss the codebook, the coding process, thematic analysis and data synthesis procedures.

After the conclusion of coding in each country, we will execute code queries for each code, stratified by respondent type (eg, providers and decision-makers) to extract code-specific data. Subsequently, we will review and summarise the code-specific and respondent-specific data from the query outputs into code summary memos using a standardised template.

Code summary memos will include a respondents table to capture brief and relevant information from each type of respondent, and narratives constructed by the researcher reviewing the query output, supported by exemplary quotes. Code summary memos will include deviant narratives and quotes that run counter to the main narrative(s) and a section for recording researcher insights on where and how codes may be connected to others. In a final step, the synthesised data in the code summary memos will be organised into thematic matrices to formalise linkages between codes and construct themes. The resulting themes will be used to report country-specific findings and to develop a refined theoretical framework. Results for each country case will be organised using the Standards for Reporting Qualitative Research checklist[75] (online supplemental file 2). To increase credibility in our findings we will consider rival explanations and triangulate across data sources (ie, SMI relevant programme documents and in-depth interviews), respondents (decision-makers and providers), researchers and social and behavioural science theories. Data collection and analysis will take place between June 2020 and June 2022.

The proposed study has several strengths including the use of theory to guide study design, data collection and reporting; the consideration of rival explanations[76]; and the use of triangulation of data sources, respondent accounts and researcher interpretation.[77] Case study research has limitations including reduced transferability of findings to other contexts and different populations of decision-makers and PHC providers.[76] Also, the use of virtual interviews may create potential loss of rapport between interviewers and respondents.

### Patient and public involvement

Neither patients nor public were involved in the conduct, reporting or dissemination of the research presented in this protocol.

## ETHICAL CONSIDERATIONS AND DISSEMINATION

The ethical approval for this study was provided by the Institutional Review Boards of the participating universities (study numbers NCR203102 for the George Washington University; SBS-2021–0162 for American University in Beirut, and P.11/20/3198 for the University of Malawi). We will follow ethical principles of voluntary and informed involvement in the study, confidentiality and safety of all participants. Verbal consent will be obtained from all respondents and be reflected in the respective interview transcripts.

A database will be maintained containing information on all interviews completed, including demographic data and time of the interview as well as confirming verbal consent by each respondent. All identifying information will be stored in an encrypted database, hosted in encrypted and password protected cloud services provided by each of the hosting research institutions. The identifier information database will be permanently deleted after the completion of data analysis.

Findings will be reported to the participating ministries of health, the commissioners of this study and to development finance partners, where applicable. Results will also be presented at local, national and international conferences and disseminated via peer-review publications. We aim to produce individual country case study manuscripts

followed by a multiple case study synthesising findings from the three study sites.

## STUDY SIGNIFICANCE

Research on the use of performance information in PHC systems is scarce; multicountry case studies in LMICs are non-existent to the best of our knowledge. The study presented here can contribute to an understanding of the contextual factors and organisational environments that enable or hinder the use of performance information in the PHC systems of El Salvador, Lebanon and Malawi. Such knowledge can inform future research and contribute to improve the strategies used in LMIC settings to measure and manage PHC system performance.

**Author affiliations**
[1]Global Health, The George Washington University Milken Institute of Public Health, Washington, District of Columbia, USA
[2]International Health, School of Health, Georgetown University, Washington, District of Columbia, USA
[3]School of Public Health and Family Medicine, University of Malawi College of Medicine, Blantyre, Malawi
[4]Obstetrics and Gynaecology, Queen Elizabeth Central Hospital, Blantyre, Malawi
[5]Department of Health Management and Policy, American University of Beirut, Beirut, Lebanon
[6]Partners In Health, Neno, Malawi

**Acknowledgements** We would like to acknowledge the Primary Healthcare Research Consortium for their support in particular Dr D Praveen and Manushi Sharma. This work was supported, in whole, by the Bill & Melinda Gates Foundation (INV-000970). Under the grant conditions of the Foundation, a Creative Commons Attribution 4.0 Generic License has already been assigned to the Author Accepted Manuscript version that might arise from this submission.

**Contributors** The study was conceptualised by WM, MM and FE-J. The first draft was written by SSW, W-CY and WM, with inputs from LD, MM and FE-J. All authors have read and approved the final manuscript.

**Funding** This study is funded by the Bill & Melinda Gates Foundation through the Primary Healthcare Research Consortium hosted at the George Institute for Global Health in India (INV-000970).

**Competing interests** None declared.

**Patient and public involvement** Patients and/or the public were not involved in the design, or conduct, or reporting, or dissemination plans of this research.

**Patient consent for publication** Not applicable.

**Provenance and peer review** Not commissioned; externally peer reviewed.

**ORCID iD**
Wolfgang Munar http://orcid.org/0000-0002-9234-987X

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
