## [Reviewer comments · BMJ Open]

ARTICLE DETAILS

TITLE (PROVISIONAL)	Characterising performance information use in the primary health care systems of El Salvador, Lebanon, and Malawi: Multiple qualitative case study protocol
AUTHORS	Munar, Wolfgang; Wahid, Syed; Makwero, Martha; El-Jardali, Fadi; Dullie, Luckson; Yang, Wen-Chien

VERSION 1 – REVIEW

REVIEWER	Lawson, Henry University of Ghana
REVIEW RETURNED	23-Feb-2022

GENERAL COMMENTS	This manuscript is written in the format of a proposal. A lot of future tense is used. There are no results, discussion, conclusion or recommendations. No limitations were discussed
---

REVIEWER	Villar-Urbe , Manuela World Bank Group
REVIEW RETURNED	01-Jun-2022

GENERAL COMMENTS	Thank you for the invitation to review this manuscript. It is an excellent idea for a study, is well written and formulated on a well developed theoretical framework. The protocol is highly relevant and timely. The selection of countries for inclusion in the study, will provide for a very useful perspective across contexts and levels of PHC measurement strategy implementation. The proposed sampling strategy and sample size across countries is likely to be sufficient but the authors should be aware of the multiple efforts and stakeholders involved in PHC measurement in Malawi and account for a potential larger sample size. I suggest that the authors more explicitly mention how interviews will include respondents that represent different actors in the data generation, processing, analysis and decision-making process; it will be important to elucidate the role of data use by those that generate the data, those that compile it and those that receive it in an aggregated manner for policy making. The authors should also consider and plan, in their data collection strategy, for a methodology to define primary health care in each country as well as performance and its measures, to ensure a comparability across respondents. The definition of PHC is very often not uniform across countries or within countries, hence a common definition at least among respondents in each country is likely to allow for comparisons and understanding of
--

	commonalities across respondents. The mentioned routine health information systems most often do not actually provide strong measures of PHC performance as no denominators are available for contrasting service delivery volumes with need; a clear definition of PHC performance measures will likely help consolidate findings across respondents. Incorporation of these abovementioned considerations will make the protocol and study stronger and more relevant.
--	--

VERSION 1 – AUTHOR RESPONSE

Reviewers' comments to Author	Author's response
Reviewer: 1. Henry Lawson University of Ghana. This manuscript is written in the format of a proposal. A lot of future tense is used. There are no results discussion, conclusion, or recommendations. No limitations were discussed	We believe that given the nature of the manuscript (the protocol of a future study), the use of future tense is appropriate. In term of sections, we followed the guidelines provided by the journal for this type of submission. We expanded the initial set of limitations to include those arising from the choice of case study design (Pages 3 and 16).
Reviewer: 2. Dr. Manuela Villar-Urbe World Bank Group The proposed sampling strategy and sample size across countries is likely to be sufficient but the authors should be aware of the multiple efforts and stakeholders involved in PHC measurement in Malawi and account for a potential larger sample size. I suggest that the authors more explicitly mention how interviews will include respondents that represent different actors in the data generation processing analysis and decision-making process; it will be important to elucidate the role of data use by those that generate the data those that compile it and those that receive it in an aggregated manner for policy making.	We agree with the recommendation to include a larger sample Malawi than in the other two countries. This decision was indeed made by our team; final sample sizes per country will be reported in the corresponding case study manuscript. Further, due to the focus of our research questions and aims, and to operational limitations (i.e., conducting research during the Covid pandemic and corresponding time constraints among respondents), this research is solely focused on the utilization of performance information by end-users such as MOH decision makers and PHC providers. A focus on data producers or data availability, while important, is outside the scope of our research.
The authors should also consider and plan in their data collection strategy for a methodology to define primary health care in each country as well as performance and its measures to ensure a comparability across respondents. The definition of PHC is very often not uniform across countries or within countries hence a common definition at least among respondents in each country is likely to allow for comparisons	We appreciate the need to provide a frame of reference and have thus included references to the definitions used by WHO (Page 4). However, given that the phenomena of interest in this study are the performance management practices used in each country and the extent to which respondents report using performance information, we will not compare their understandings about what PHC is. In our opinion, the

and understanding of commonalities across respondents.	latter would constitute a separate research question. We will, however, inquire as to the respondents' knowledge of and opinions about each ministry's PHC priorities as these pertain to service delivery routines (by providers) and managerial practices (among MOH decision makers). Finally, aware of the current debate about decolonizing global health, we believe that we are in no position to advocate for, or 'push,' towards universal definitions of PHC.
The mentioned routine health information systems most often do not actually provide strong measures of PHC performance as no denominators are available for contrasting service delivery volumes with need; a clear definition of PHC performance measures will likely help consolidate findings across respondents.	We agree that RHUS may not necessarily be optimal sources of PHC system performance. However, they are widely used by governments and oftentimes they are the sole available source of information to monitor PHC system performance. Other data sources such as population and/or facility surveys tend to be donor-driven and are less common than in OECD countries. While we intend to explore the use of performance information from all available sources, characterizing specific PHC performance measures in use in each country is outside of the scope of this research.

VERSION 2 – REVIEW

REVIEWER	Villar-Urbe , Manuela World Bank Group
REVIEW RETURNED	03-Oct-2022

GENERAL COMMENTS	The ideas presented in this protocol and results of the proposed study will be of great value to those working to support or improve PHC performance measurement in countries across the world. I appreciate the efforts that the authors have made to address my comments to the previous draft; (i) openness to increase the sample size in Malawi is welcome, (ii) further clarity on the intent to interview only public sector care providers and MOH staff is also helpful (although it provides a limited view of the PHC system, regardless of definition that the authors might choose to highlight), (iii) introduction of a WHO definition of PHC, to the text, and finally (iv) the clarification that RHIS information might be complemented with other information to provide a more complete picture of the PHC system's performance and understanding of patterns for its utilization.
---